# Identification of a New Drug Binding Site in the RNA-Dependent-RNA-Polymerase (RdRp) Domain

**Aparna S. Gana** [1,*] **and James N. Baraniuk** [2,*]

[1] College of Arts and Sciences, University of Virginia, 402 Balz Dobie, Charlottesville, VA 22904-3104, USA
[2] Division of Rheumatology, Immunology and Allergy, Department of Medicine, Georgetown University Medical Center, 3900 Reservoir Rd NW, Washington, DC 20007, USA
[*] Correspondence: qfx4yk@virginia.edu (A.S.G.); baraniuj@georgetown.edu (J.N.B.)

**Abstract:** We hypothesize that in silico structural biology approaches can discover novel drug binding sites for RNA-dependent-RNA-polymerases (RdRp) of positive sense single-strand RNA (ss(+)RNA) virus species. RdRps have a structurally conserved active site with seven motifs (A to G), despite low sequence similarity. We refined this architecture further to describe a conserved structural domain consisting of motifs A, B, C and F. These motifs were used to realign 24 RdRp structures in an innovative manner to search for novel drug binding sites. The aligned motifs from the enzymes were then docked with 833 FDA-approved drugs (Set 1) and 85 FDA-approved antivirals (Set 2) using the Molecular Operating Environment (MOE) docking 2020.09 software. Sirolimus (rapamycin), an immunosuppressant that targets the mammalian mTOR pathway, was one of the top ten drugs for all 24 RdRp proteins. The sirolimus docking site was in the nucleotide triphosphate entry tunnel between motifs A and F but distinct from the active site in motif C. This original finding supports our hypothesis that structural biology approaches based on RdRp motifs that are conserved across evolution can define new drug binding locations and infer potential broad-spectrum inhibitors for SARS-CoV-2 and other ss(+)RNA viruses.

**Keywords:** RNA-dependent-RNA-polymerase (RdRp); sirolimus; rapamycin; single-stranded positive-sense RNA viruses; ss(+)RNA; evolutionarily conserved region; common core; drug design; Molecular Operating Environment (MOE)





## 1. Introduction

We hypothesize that in silico structural biology approaches can discover novel drug binding sites for RNA-dependent-RNA-polymerases (RdRp) of positive single strand RNA (ss(+)RNA) virus species.

SARS-CoV-2, the causative agent of COVID-19, is responsible for the death of 6 million people worldwide. Mobilization of a strong scientific response has resulted in 358,177 peer reviewed publications on COVID-19 as of 14 June 2023 in LitCovid, a comprehensive hub for tracking up-to-date scientific publications [1]. About 54,441 publications have applied in silico methods to search for drugs that inhibit SARS-CoV-2 enzymes. These efforts have identified many potential therapeutics and have led to the repurposing of FDA antiviral drugs such as remdesivir which is an Ebola RdRp inhibitor. However, remdesivir has had mixed benefits for clinical improvement and mortality compared to placebo [2–4]. The introduction of effective vaccines may have stemmed the tide so that COVID-19 enters an endemic stage, but there is still a need for more specific and effective antiviral drugs. The pandemic exposes the risk that dengue, West Nile, Zika, Ebola and currently understudied zoonotic viruses may threaten global health. So far, drug design efforts have focused on individual viruses. We propose an alternative approach by applying broad drug-design principles against essential enzymes that share evolutionary origins and conserved structural domains across a host of viruses. Nucleic acid polymerases that

include RNA-dependent-RNA-polymerase (RdRp), DNA-dependent-RNA-polymerase (DdRp), DNA-dependent-DNA-polymerase (DdDp) and reverse transcriptases are good candidates for such an approach [5].

The rationale for pursuing novel drug binding studies for RdRp is that this enzyme is highly conserved by ss(+)RNA viruses [6]. The primacy of RdRp in viral evolution is epitomized by the simplest of virus families, the Narnaviridae, which is subdivided into two genera: Narnavirus and Mitovirus [7]. Narnaviridae consists of a single naked RNA strand without a capsid. The 2.5 kB to 2.9 kB genome codes only for RdRp and no structural, capsid or enzymatic proteins. Its RdRp is 80 to 104 kDa and is structurally related to other RdRp enzymes including those of the Leviviridae RNA bacteriophages [8]. RNA viruses were transferred from a proto-mitochondrial prokaryote into a proto-eukaryotic host as part of mitochondrial endosymbiosis [9]. Subsequent horizontal gene transfer of other viral and host genes and gene duplication likely led to the growth, evolution and divergence of *Picornaviridae* (Rhinovirus), *Flaviviridae* (West Nile virus), *Filoviridae* (Ebola virus), *Coronaviridae* (Coronavirus) and other families (Figure 1) [10]. Coronaviruses have the largest genomes (27.6 to 31 kB) [11]. An alternative hypothesis is that Narnaviridae evolved by loss of genes and proteins.

RdRp folds into a right-hand architecture with palm, fingers, and thumb domains for template binding, nucleoside triphosphate (NTP) entry, active site, polymerization and associated functions [12,13]. Structural superposition of all ss(+)RNA virus RdRp structures from the Protein Data Bank (PDB) identified conserved motifs with finger tips curling up towards the thumb to form an arch over the palm and its active site [10]. The fingers and thumb create tunnels for template RNA entry, outlet for the duplex RNA exit and transit of nucleotide triphosphate (NTP) to the active site and subsequent exit of pyrophosphate. RdRp harbors seven conserved structural motifs that are arranged in the order G, F, A, B, C, D and E from the N-terminus to the C-terminus of the proteins [10,14,15]. Motifs F, A, B and C are highly conserved across all ss(+)RNA viruses, indicating their pivotal role in catalysis despite extensive amino acid divergence over the course of evolution [16]. We refer to this as the shared conserved RdRp "motif FABC". Motifs D, E and G exist in spatial positions around the active site but have become structurally diverged among ss(+)RNA viruses. Current RdRp drugs bind to the active site, which is positioned within Motif C [17,18]. Current drug design efforts have been focused on one virus at a time, but we propose a wider search based on the shared motif FABC.

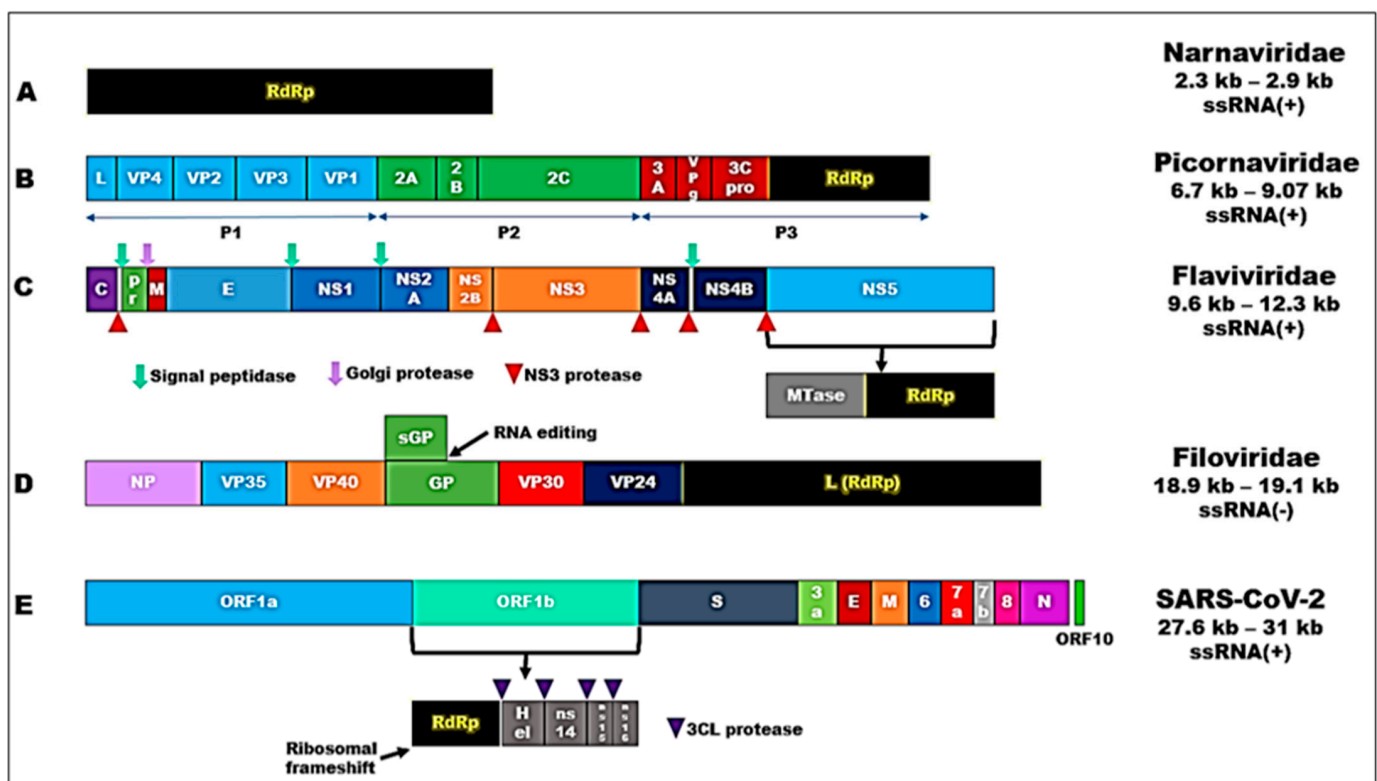

**Figure 1.** Simplified genome architectures of RNA virus families, (**A**) Narnavirirdae, (**B**) Picornaviridae, (**C**) Flaviviridae, (**D**) Filoviridae and (**E**) Coronaviridae, ordered from smallest to largest. The RdRp domain was conserved among all families. Narnaviridae contain only a single ORF (open reading frame) encoding RdRp. Flaviviridae encode a large non-structural protein 5 (NS5) with RdRp and methyltransferase (MTase) domains that facilitate viral replication [19]. Filoviridae encode the L-protein, which contains the RdRp domain [20]. Coronaviridae encode ORF1b which contains the RdRp domain (NSP12), helicase (NSP13), two ribonucleases (NSP14, NSP15) and RNA-cap methyltransferases (NSP14, NSP16) [21]. Genome architectures were made in PowerPoint using *ViralZone* as a reference (https://viralzone.expasy.org/, accessed on 1 April 2022).

Factors that have limited efforts to discover broad spectrum antivirals include the high viral mutation rate [22] and extreme sequence divergence [12] between species [23,24]. We address this problem by relying on the structurally conserved motif FABC. Key invariant amino acids in these motifs are vital for replication and so are unlikely to harbor mutations that would lead to drug resistance [25]. We used the shared motif FABC to create RdRp structures that would share potential novel drug binding sites that were conserved across all ss(+)RNA viruses. Our drug docking search identified a site between motifs F and A that was spatially distant from Motif B and the active site in Motif C. To our knowledge this is the first search for drug binding sites that may be conserved across a wide array of ss(+)RNA viruses and that does not involve the active site.

The high structural conservation of RdRp and other polymerases generates the additional hypothesis that other monomeric viral polymerases including those of dsDNA and ss(-)RNA strand viruses descend from a common ancestor [26,27] and suggests that the shared structural motif approach may be extended to other polymerase families [10].

## 2. Results

### 2.1. RdRp Motifs

The "common core" for the active site of ss(+)RNA virus RdRps is defined by structural motifs A to G [10]. Motifs A, B and C form the palm of the right-handed structure with motif F in the fingers (Figures 2 and 3). Motifs D and E in the palm and G in the fingers are

positioned in approximately the same three-dimensional space across species but cannot be exactly superimposed (Figure 2B). The thumb does not contribute to the active site. In general, the fingers interact with the major groove of the template RNA.

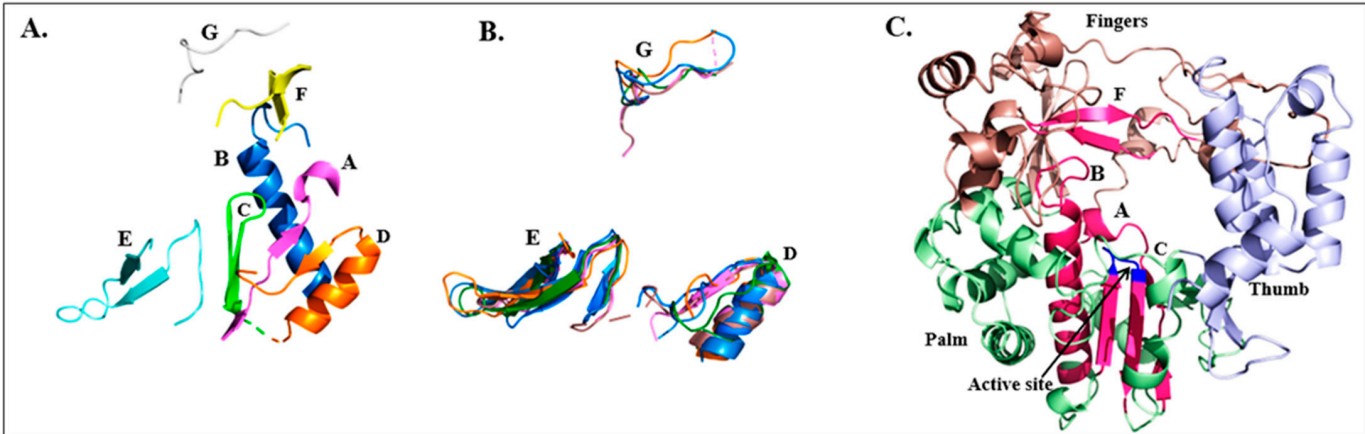

**Figure 2.** RdRp motifs. (**A**) Conserved sequence motifs A to G were extracted from poliovirus (PDB ID: 2ijf) and superimposed. Motifs A, B, C and F defined by Peersen [10] were highly conserved across ss(+)RNA viruses (Figure 3). (**B**) Motifs D, E and G were not highly conserved when representative structures of each family were superpositioned: hepatitis C virus (gold; PDB ID: 1c2p), Thosea asigna virus (blue; PDB ID: 4xhi), SARS-CoV-2 (green; PDB ID: 7bw4), enterovirus D68 (pink; PDB ID: 6l4r) and Norwalk virus (light blue; PDB ID: 2b43). The alpha helix of motif D was well aligned but the contiguous loop that follows deviated extensively. There was little structural conservation in motif G. (**C**) Cartoon diagram of poliovirus RdRp (PDB ID: 2ijf) shows the palm (pale green), fingers (light brown) and thumb (light purple) domains, and motifs A, B, C and F that form the conserved motif FABC (magenta) in this manuscript. The active site (dark blue, black arrow) is at the tip of the motif C. Figure created using PyMol 2.5.

**Table 1.** ssRNA(+) viruses chosen for our study.

| Picornaviridae | Flaviviridae | Caliciviridae | Coronaviridae | Permutotetraviridae |
|---|---|---|---|---|
| • Encephalomyocarditis virus (EMCV) PDB: 4nz0 <br> • Coxsackievirus B (COXB) PDB: 3ddk <br> • Enterovirus 71 (ev71) PDB: 3n6l <br> • Enterovirus D68 (EV-D68) PDB: 6l4r <br> • Poliovirus (POLI) PDB: 2ijf <br> • Porcine aichi virus PDB: 6R1I <br> • Sicinivirus A PDB: 6QWT <br> • Rhinovirus (RHIN) PDB: 1tp7 <br> • Foot-and-mouth disease virus (FMDV) PDB: 1u09 | • Dengue virus (DENV) PDB: 2j7u <br> • Tick-borne encephalitis virus (TBEV) PDB: 7d6n <br> • Zika virus (ZIKV) PDB: 5wz3 <br> • Yellow fever virus (YFV) PDB: 6qsn <br> • West Nile virus (WNV) PDB: 2hcn <br> • Japanese encephalitis virus (JEV) PDB: 4hdh <br> • Bovine viral diarrhea virus (BVDV) PDB: 2cjq <br> • Classical swine fever virus (CSFV) PDB: 5yf5 <br> • Hepatitis C virus (HCV) PDB: 1c2p | • Norwalk virus (human norovirus) PDB: 2b43 <br> • Rabbit Hemorrhagic Disease Virus (RHDV) PDB: 1khv <br> • Sapporo virus PDB: 2ckw <br> • Murine norovirus (MNV) PDB: 3uqs | • Severe acute respiratory syndrome coronavirus 2 (SARS-CoV-2) PDB: 7bw4 | • Thosea asigna virus (TAVP) PDB: 4xhi |

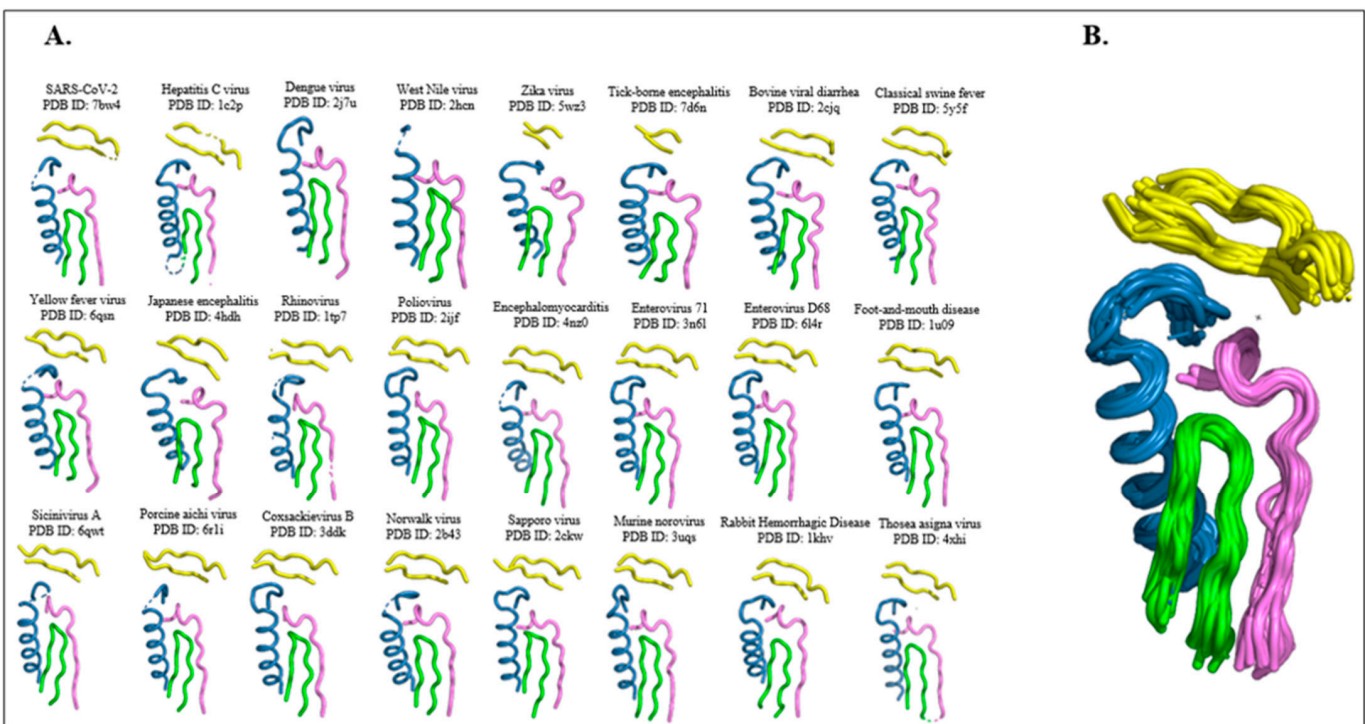

**Figure 3.** Shared conserved RdRp motif of ss(+)RNA viruses. (**A**) The evolutionarily conserved RdRp common core was identified for the 24 ss(+)RNA viruses listed in Table 1. The core contains structural motifs A (pink) and C (green) in the active site which are conserved among monomeric polymerase classes, motif F (yellow), which is conserved across most RdRp domains, and motif B (blue), which is conserved in ss(+)RNA viruses [10]. Motif F was absent or shortened in the Flaviviruses including dengue virus, West Nile virus, Zika virus and tick-borne encephalitis virus, but the binding site was maintained by analogous sequences of the protein. (**B**) Structural alignment of the 24 motif FABC structures showing the highly conserved architecture. Figure was created with the PyMol Visualization software (https://pymol.org/2/).

Each motif has a specific function to position RNA and an incoming NTP for replication. Motifs A, B and C in the palm form the floor of the active site (Figure 2). Motif B locks the RNA template ribose in place from below, opposite motif F that arches over top of the active site. Aspartates in motif A interact with the metal ion of the incoming nucleotide's phosphates and ribose-3′-OH. The second aspartate and an asparagine in Motif B discriminate against deoxynucleotides [12]. Motif C has a pair of aspartate residues that form the active site by binding a metal ion that facilitates the ribose–phosphate ligation reaction. Motif D is a mobile unit for chain elongation and one lysine aids release of pyrophosphate. Motif E at the palm–thumb junction provides a "primer grip" for correct positioning of the 3′-OH for catalysis. Motif F in the fingers plays a crucial role in setting the correct geometry of the RNA template strand and locking its target nucleotide in place for recognition by Watson–Crick binding. Motif F has two antiparallel strands of a hairpin structure linked by a loop. The strands are conserved but the loop varies between species. The loop is important for drug binding. Motif F is truncated or lost in Flaviviridae including dengue, Zika, West Nile and tick-borne encephalitis viruses. Structural evolution in Thosea asigna has led to motif swapping where the order is F, C, A and B in the protein instead of the usual F, A, B and C (Figure 4). Motif G contributes to the template entrance channel.

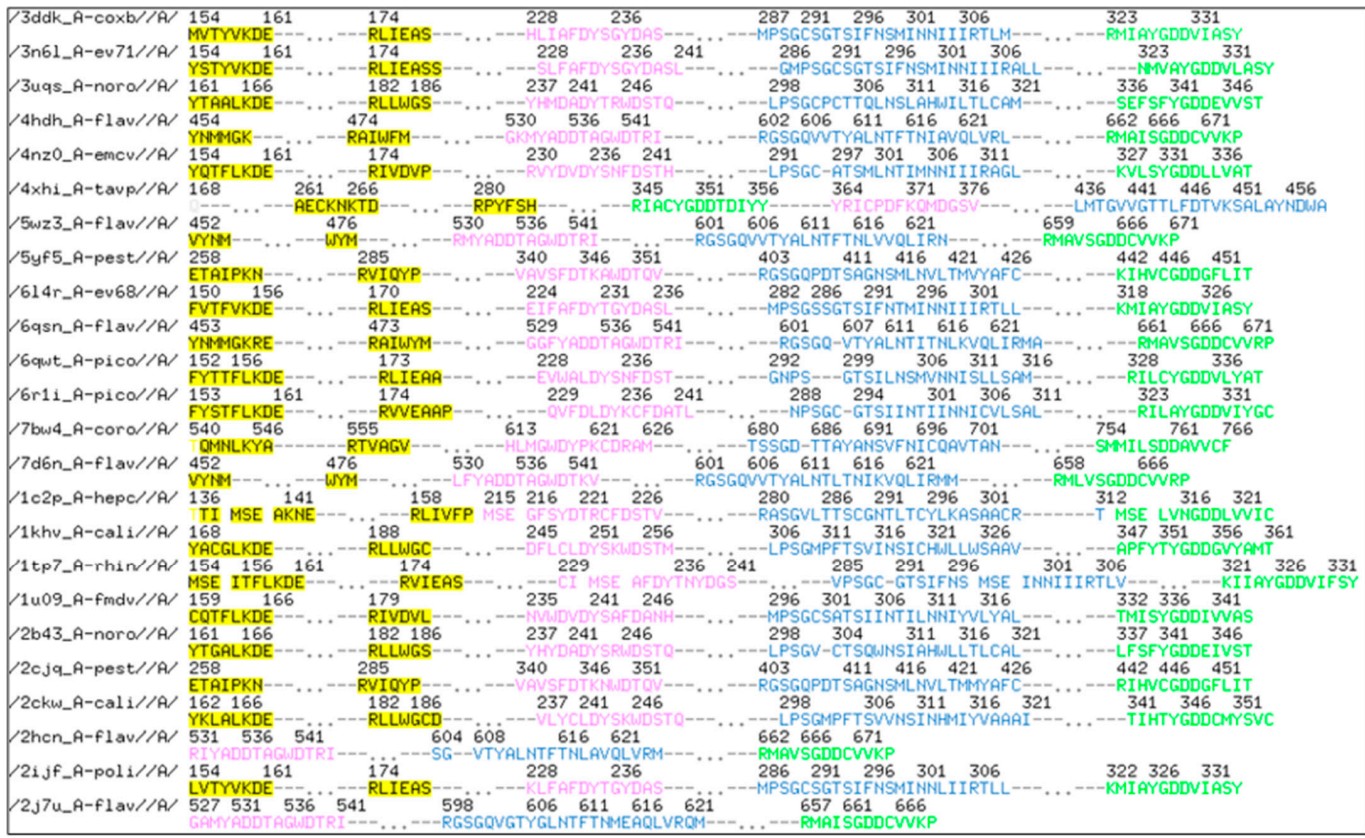

**Figure 4.** Amino acid sequences of shared structural motifs. Structural motifs F (yellow), A (pink), B (blue) and C (green) are shown. Figure was created with the PyMol Visualization Software.

We re-aligned 24 representative RdRp structures (Table 1) based on motifs A to G using PyMOL Visualization software (https://pymol.org/2/) and found a more highly conserved, structurally aligned motif consisting of motifs F, A, B and C (Figure 3B) that we called the shared conserved RdRp motif.

The amino acid sequence alignment of motifs F, A, B and C is shown in Figure 4 with intervening nonconserved sequences deleted for clarity. The two antiparallel strands of motif F (yellow) are shown without the intervening loop that connects them.

Consensus conserved residues in the aligned sequences were analyzed using Muscle multiple sequence online tool (https://www.ebi.ac.uk/Tools/msa/muscle/, accessed on 21 May 2022) with SARS-CoV-2 (PDB 7bw4 chain A) as the reference for sequence numbering (Figure 5). Several novel observations about sequence homology were made. Eight amino acids are 100% conserved among the ss(+)RNA viruses. Their positions may be stabilized by adjacent nonpolar amino acids (top row, green) that interact with nonconserved parts of the protein. Motif A contains D618xxxxD623 with Y or T in position 619 in all 24 viruses and is likely essential for ferrying the incoming nucleotide triphosphate into the active site. Motif B has a loop and an alpha helix with conserved S682 G683, T686, N690, S/T691 and N/H694. Motif C has two beta strands with a hairpin loop of G759 D760 D761uhh where u is uncharged, and h is hydrophobic. The two aspartates of the GDD sequence form the active site. However, SARS-Cov2 has the sequence of SDD (instead of G) which is also found in negative stranded and double-stranded RNA polymerases. We propose the SDD sequence may decrease the effectiveness of polymerase inhibitors designed for sequences with GDD. When the sequences of the two strands of motif F were condensed, they show a conserved KxxR (K544xx-break-R555) pattern in all viruses except flaviviruses where motif F is truncated or absent. We searched the flavivirus structures, and propose that R222 from an adjacent strand is positioned close to the binding site and may fulfill the function

of motif F. Over half of the viruses have the sequence KDER in motif F and the sequence N690xxxN694 in motif B.

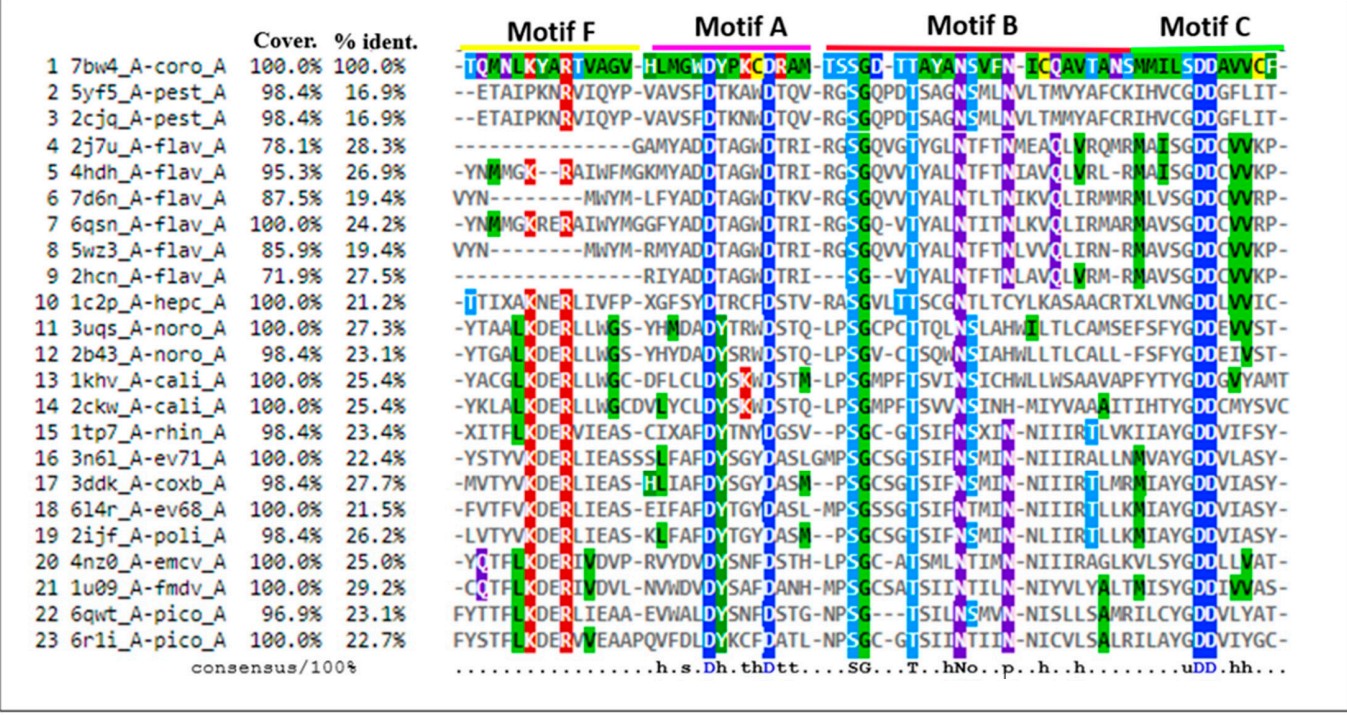

**Figure 5.** Sequence based alignment of shared conserved RdRp motif. Multiple sequence alignment was performed for motifs F (yellow bar), A (magenta), B (red) and C (green) using the Muscle online alignment tool with SARS CoV-2 PDB 7bw4_A-coro_A as the reference (top row). Conserved residues from the motifs were condensed into linear format by removing non-conserved intervening sequences. The motif F sequence contains the two strands with the loop removed (Figure 4). The importance of the motif structures was apparent from the high conservation of key residues and % coverage. Sequence diversity was shown by low percent identity. Conserved residues are color coded as red for positively charged, blue for negatively charged, green for hydrophobic, purple for polar and light blue for small polar residues. The bottom row indicates consensus hydrophobic residues by h, p for polar, t for turn-like, s for small, o for alcohol and u for tiny.

Structural relationships between the 24 RdRps were assessed by neighbor joining phylogenetic trees using parameters $Q_H$ (a metric for measuring structural homology) and Root Mean Squared Deviation (RMSD) implemented in the MultiSeq application within VMD [28] (Figure 6). Coronavirus and Thosea asigna were most diverged from the other viruses by both $Q_H$ and RMSD. Picornaviridae formed the largest clade. The Flaviviridae were divided into two subclades with Hepatitis C, classical swine fever and bovine viral diarrhea viruses separated from dengue, West Nile, Yellow Fever and other flaviviruses. Structural homologies within each clade may offer additional opportunities for future drug development. This supports our overall hypothesis.

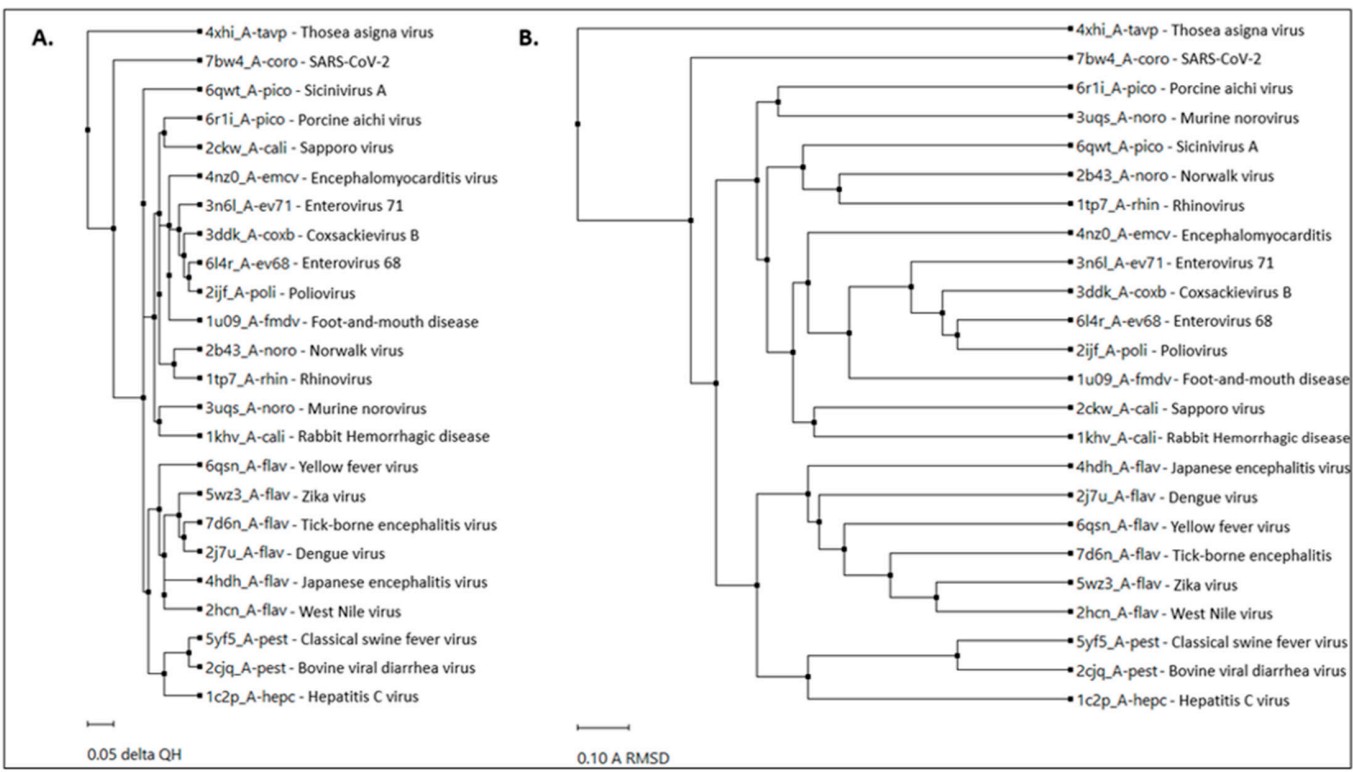

**Figure 6.** Phylogenetic relationships for the shared conserved motif. Structural trees were based on (**A**) $Q_H$ and (**B**) RMSD. Both phylogenetic trees were created using the MultiSeq routine in the Visual Molecular Dynamics (VMD) software [29].

### 2.2. Drug Docking

Our hypothesis predicted that drugs would bind to novel motifs shared by all RdRps when aligned based on the conserved motif FABC (Figure 3B). Two sets were tested. Set 1 was 833 FDA-approved drugs and Set 2 was 85 FDA-approved antiviral drugs. The 24 RdRp structures aligned according to motif FABC were individually docked with each drug using Molecular Operating Environment (MOE) docking 2020.09. The top ten drugs with the lowest docking scores (S-scores, kCal/mole) were selected for each virus. The S-score is calculated based on the number of electrostatic and non-electrostatic interactions and gives a prediction of binding affinity between drug and protein. Very negative scores indicate strong binding and suggest favorable drug binding. The top 10 drug hits with highest affinity for each RdRp were tabulated, then the most frequent drugs enumerated.

Sirolimus was in the top 10 hits for all 24 viruses (Figure 7) followed by tacrolimus (*n* = 22) and pimecrolimus (*n* = 19). Four rifampin antibiotics had 8 to 18 hits each. Digoxin was next with 17 hits. The results suggest that -limus drugs, rifampin and digoxin may be viable platforms for drug development. Supplementary Material Table S1 contains detailed information about the docking results (e.g., S-score values) and drug properties for the top ten drugs for each virus.

Antiviral drugs (Set 2) had lower affinities for the shared structure (Supplementary Material Table S2), and much lower S-scores compared to Set 1 drugs. The top hits for antivirals were asunaprevir and faldaprevir with 21 hits each (Figure 8). They inhibit NS3/4A protease of Hepatitis C. Elbasvir, pibrenstasvir, velpatsavir (20 hits each) and daclatasvir (17 hits) are HepC NS5A protease inhibitors.

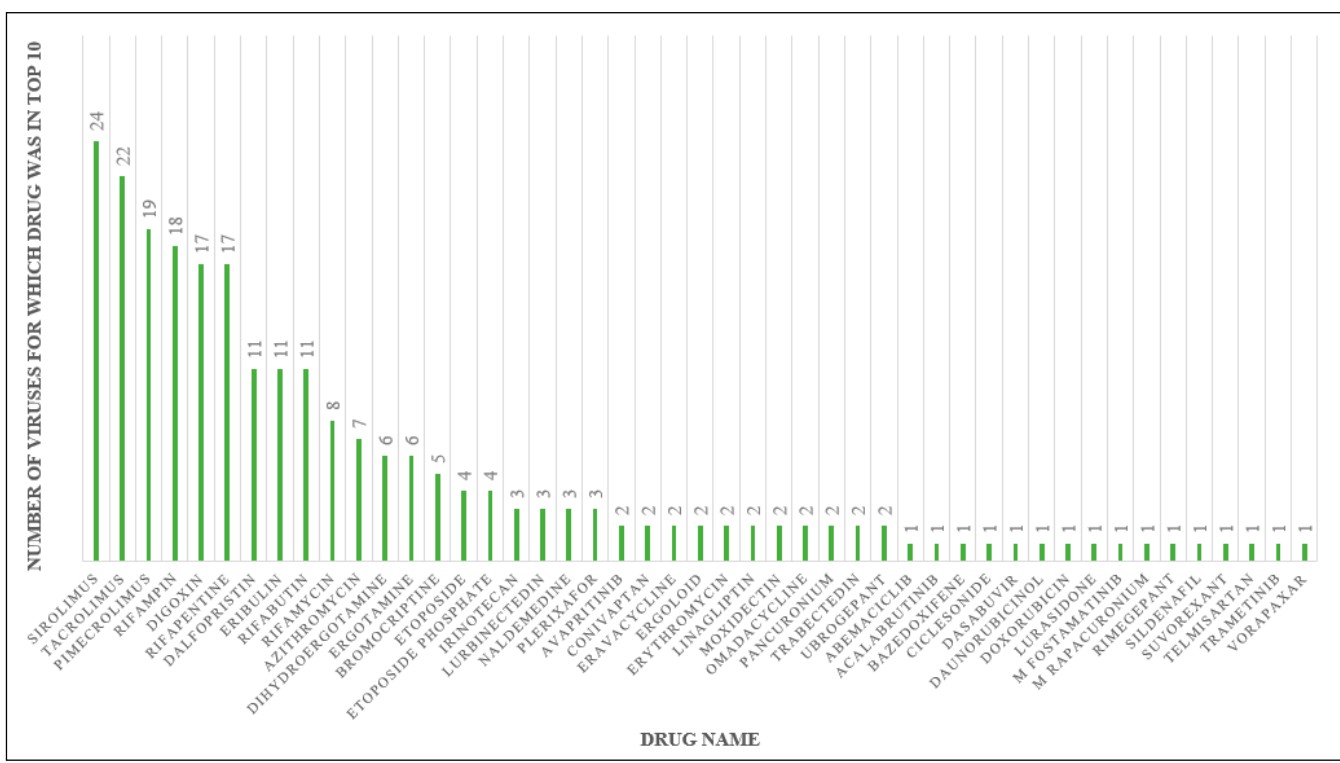

**Figure 7.** Set 1 drugs. The number of viruses for which each Set 1 drug was in the top ten. Sirolimus was in the top 10 for all 24 viruses.

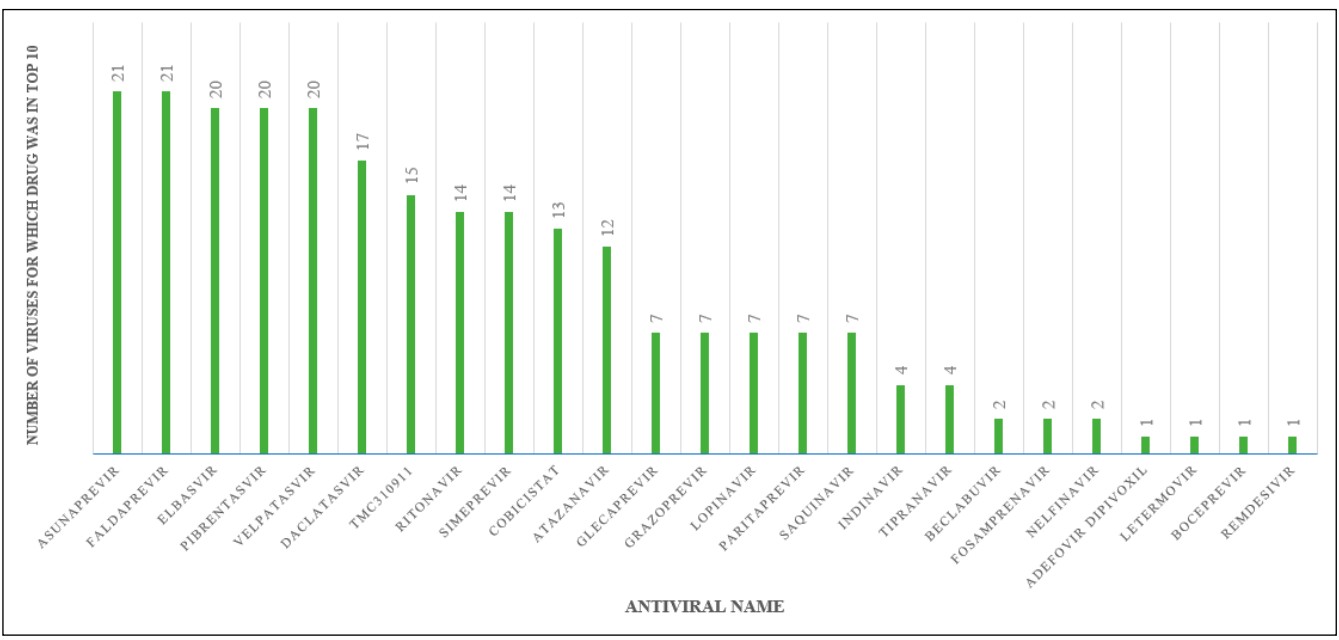

**Figure 8.** Set 2 antiviral drugs. The top matches were inhibitors of hepatitis C proteases NS3, NS3/4A and NS5A.

Remdesivir is a delayed chain terminator active at RdRp but was in the top ten for only one virus (Picornaviridae—Enterovirus 71) indicating its low binding to the shared conserved motif of ss(+)RNA RdRps. Remdesivir was developed as a NS5B polymerase inhibitor for hepatitis C and has been studied for Ebola, Marburg, respiratory syncytial virus and COVID-19. Other nucleoside analogue inhibitors, nucleoside/nucleotide reverse

transcriptase inhibitors (HIV), and non-nucleoside analogue inhibitors of RdRp that may act allosterically outside the hepatitis C active site were not identified [30].

*2.3. Sirolimus Binding Site*

The prototypic sirolimus binding site was located adjacent to the active site between motifs A and F (Figure 9). The orientation of sirolimus was different in each RdRp (Figure 10). Most of the interactions were with motifs A, F and residues in the loop connecting the two strands of motif F. There was no interaction with the aspartate dimer in motif C that mediates nucleotide ligation to the growing RNA chain. Motif B had no sirolimus binding. The binding mode of sirolimus is very different from that discussed in a recent paper by Piplani et al. [17], where sirolimus was matched to the active site only, where it interacted with Asp761. That mode of binding was not found in our model.

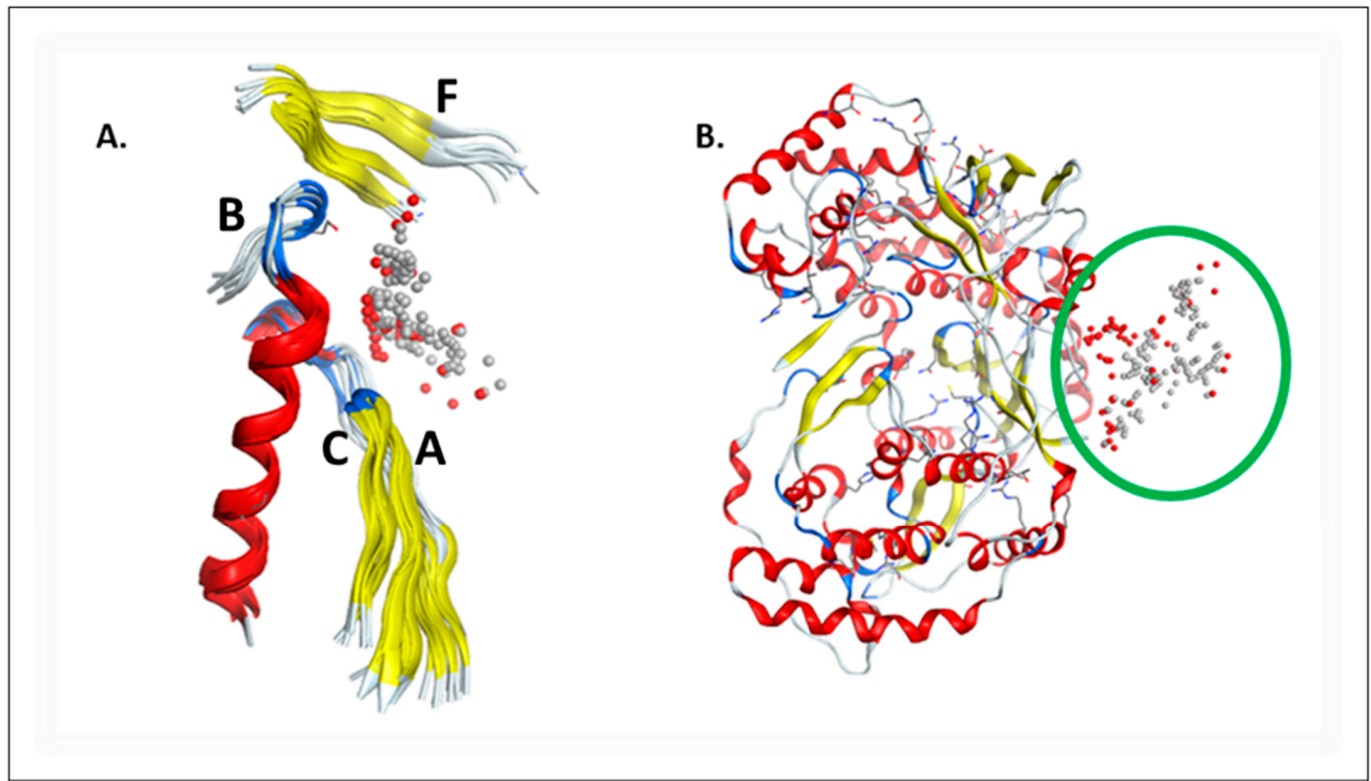

**Figure 9.** Drug docking site adjacent to the active site of RdRp. (**A**) The shared motifs A, B, C and F that were conserved in RdRp and the drug docking site are shown with red and white balls to indicate atomic centers. (**B**) The docking site (green circle) is displayed over the structure of Hepatitis C (PDB ID: 1c2p) RdRp with alpha helices in red, beta strands in yellow, and loops in grey.

The sirolimus binding site was investigated by docking the drug to the structure of poliovirus RdRp (PDB ID: 2ijf) in complex with RNA (PDB ID: 3ol6). Sirolimus bound to the exterior of RdRp over the NTP entry tunnel (Figure 11A). It interacted with NTP binding residues in motif A and the folded back N-terminal domain of RdRp (Figure 11B) that has a significant influence on catalysis and fidelity during RNA synthesis. The RNA strands did not obstruct the sirolimus binding site. Sirolimus did not interact with the active site (Figure 11C) or NiRAN (Nidovirus RdRp associated nucleotidyl transferase) domain (PDB IDs-7BTF and 6M71) [31]. This analysis indicates sirolimus bound to the NTP entry tunnel outside the active site and would sterically obstruct access of NTP to motif C for nucleotide recognition and chain ligation and elongation. Alternatively, drugs in this location may generate allostatic modulation or interfere with motif rotation during ligation and template RNA advancement.

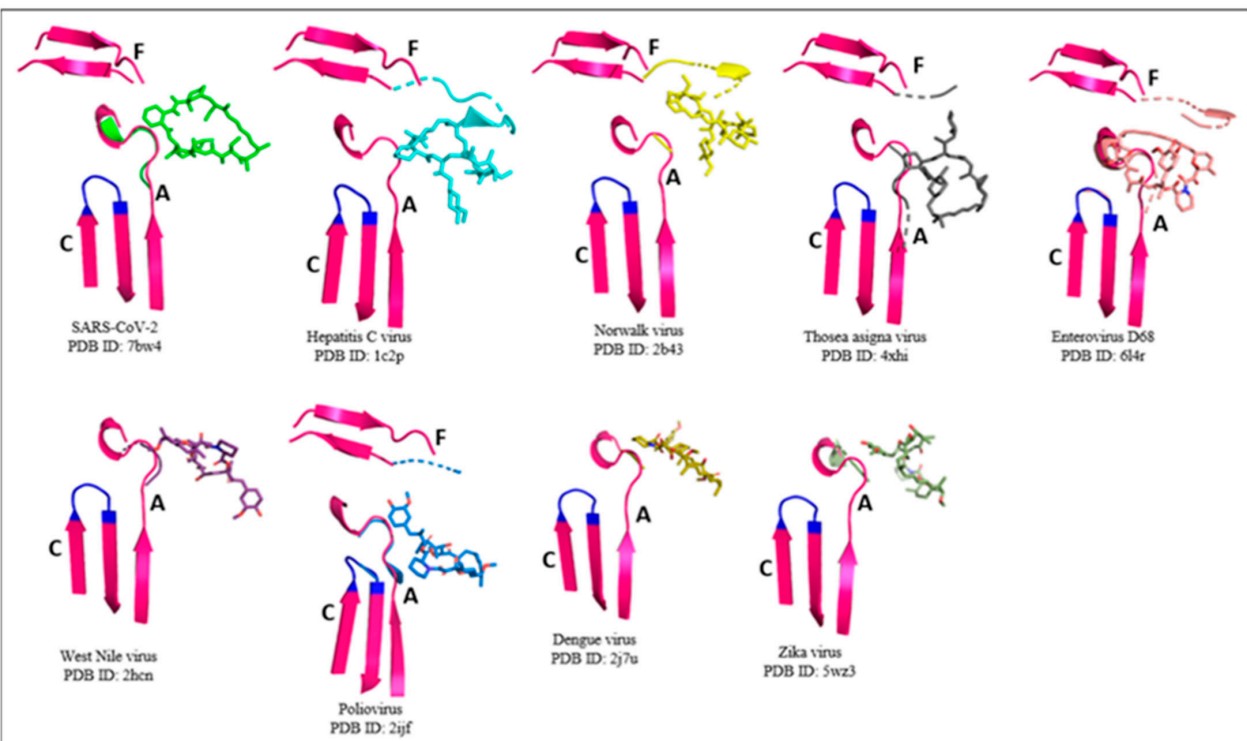

**Figure 10.** Sirolimus binding to conserved motifs F, A and C. Sirolimus interacts with either motif A in the palm or motif F in the fingers, or both, for these representative RdRps. In Enterovirus D68 and Poliovirus, sirolimus interacts with motif C in the active site (blue) in addition to motifs A and F. Motif B does not participate in any interaction with sirolimus and hence was not included in the figure. The NTP tunnel, other motifs and sequences were not shown for clarity, but also contributed to the sirolimus binding site. The figure was created with the PyMol Visualization Software.

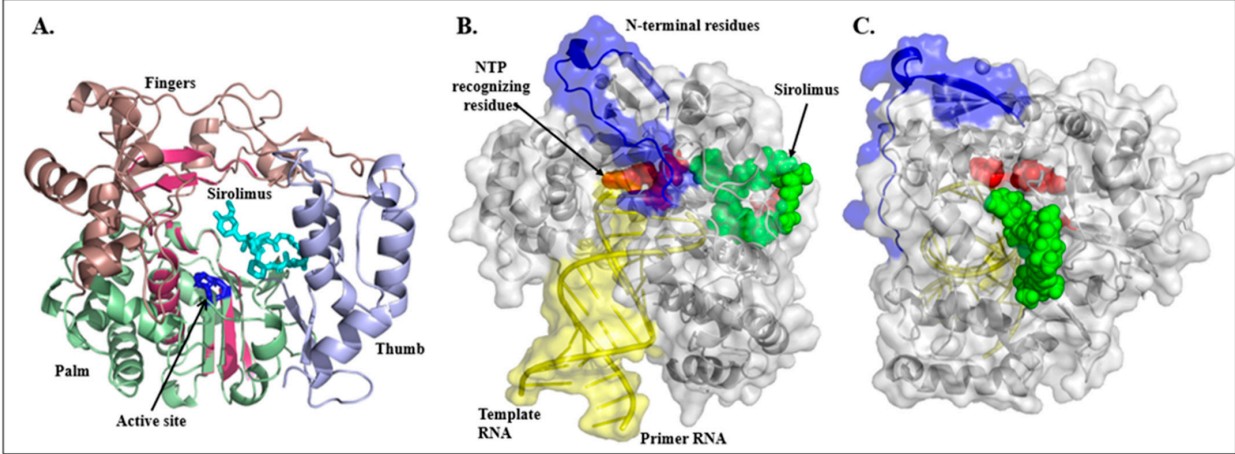

**Figure 11.** Sirolimus and Poliovirus RdRp. (**A**) This cartoon diagram of poliovirus RdRp (PDB ID: 2ijf) shows the palm (pale green), fingers (light brown) and thumb (light purple) domains, motifs in the common core (magenta) and the active site (dark blue). Sirolimus (cyan) is bound adjacent to the active site (dark blue). (**B**) In Poliovirus RdRp (PDB ID: 3ol6), sirolimus (green spheres) was bound by residues in the N-terminal (blue) and NTP entry tunnel that were adjacent to but did not overlap with active site residues D233, D238, S288, G289, N297 and D328 (red spheres) that take part in NTP recognition and catalysis. The exit tunnel for the template—nascent RNA duplex (yellow) is shown for orientation. (**C**) 90-degree rotated view of PDB ID: 3ol6 shows the separation between the sirolimus binding site and active site (red spheres). Figure created with PyMol visualization software.

Sirolimus interacted with several basic amino acids (Figure 12) that predicted strong electrostatic interactions between drug and protein. The electrostatic surface was examined by docking sirolimus to the structure of SARS-CoV2 RdRp with template and primer RNA and remdesivir in the active site (PDB ID: 7BV2) (Figure 12). The template RNA entry groove and duplex RNA exit tunnel were basically charged to facilitate electrostatic interactions with the phosphate backbones for RNA translocation through the machine. Motif C contributed catalytic aspartate residues to the acidic active site. Remdesivir was bound to the primer RNA in the active site next to motif C. The NTP entry tunnel was basically charged in order to promote transit and exchange of NTP and its pyrophosphate and metal ion before and after ligation. Sirolimus did not bind to the active site and did not overlap with remdesivir. These studies in coronavirus and poliovirus confirmed that the sirolumus binding pocket was between motif A, motif F and its intervening loop at the NTP entry tunnel and did not involve the catalytic site on motif C.

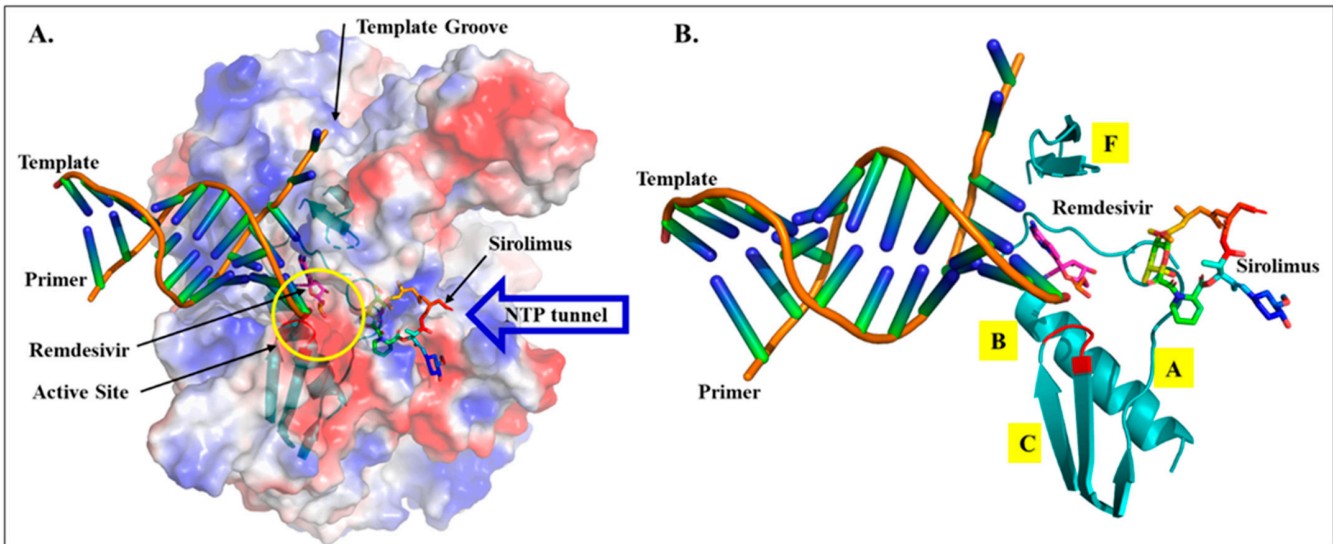

**Figure 12.** Sirolimus and remdesivir interactions. (**A**) Electrostatic view of the SARS-CoV2 RdRp with remdesivir (PDB ID: 7BV2) with basic (blue) and acidic (pink) surfaces. The phosphate backbone of the template RNA in the template entry groove and dsRNA in the exit tunnel interact with the positively charged surface. The active site (yellow circle) is acidic (pink) from aspartates in motifs A and C. Sirolimus interacts with a basically charged surface in the NTP entry tunnel (blue arrow) adjacent to the active site but does not overlap with the remdesivir binding pocket. For clarity the thumb, magnesium ions and pyrophosphate have been removed. (**B**) Same view as (**A**) without the electrostatic surface showing the shared conserved RdRp motifs (A, B, C, F), remdesivir in the active site with motif C and dsRNA and sirolimus between motifs A and F blocking the NTP entry tunnel. This figure was created with the PyMol visualization software.

The electrostatic nature of NTP entry tunnel and active has been stylized by negatively charged acidic motif C aspartates 760 and 761 that interact with the $Mg^{2+}$ ions, the triphosphate string and the basic positively charge surface of R553 and R555 from motif F, K621 in motif A and K798 that is distal to motif D [32].

RdRps have open spaces that form thumb pockets I (T1) and II (T2) and palm pockets I (P1), II (P2) and β (Pβ) that are adjacent to the active site [12]. The adjacent pockets form channels or tunnels that allow entry of the RNA template to the heart of the enzyme, exit of the product strands and access for NTP, metal ions, pyrophosphate and drugs to traverse the interior of the enzyme during replication. The entry and exit tunnels are lined with positively charged amino acids to facilitate these interactions. The sirolimus binding site was at the NTP entry tunnel (Figure 13).

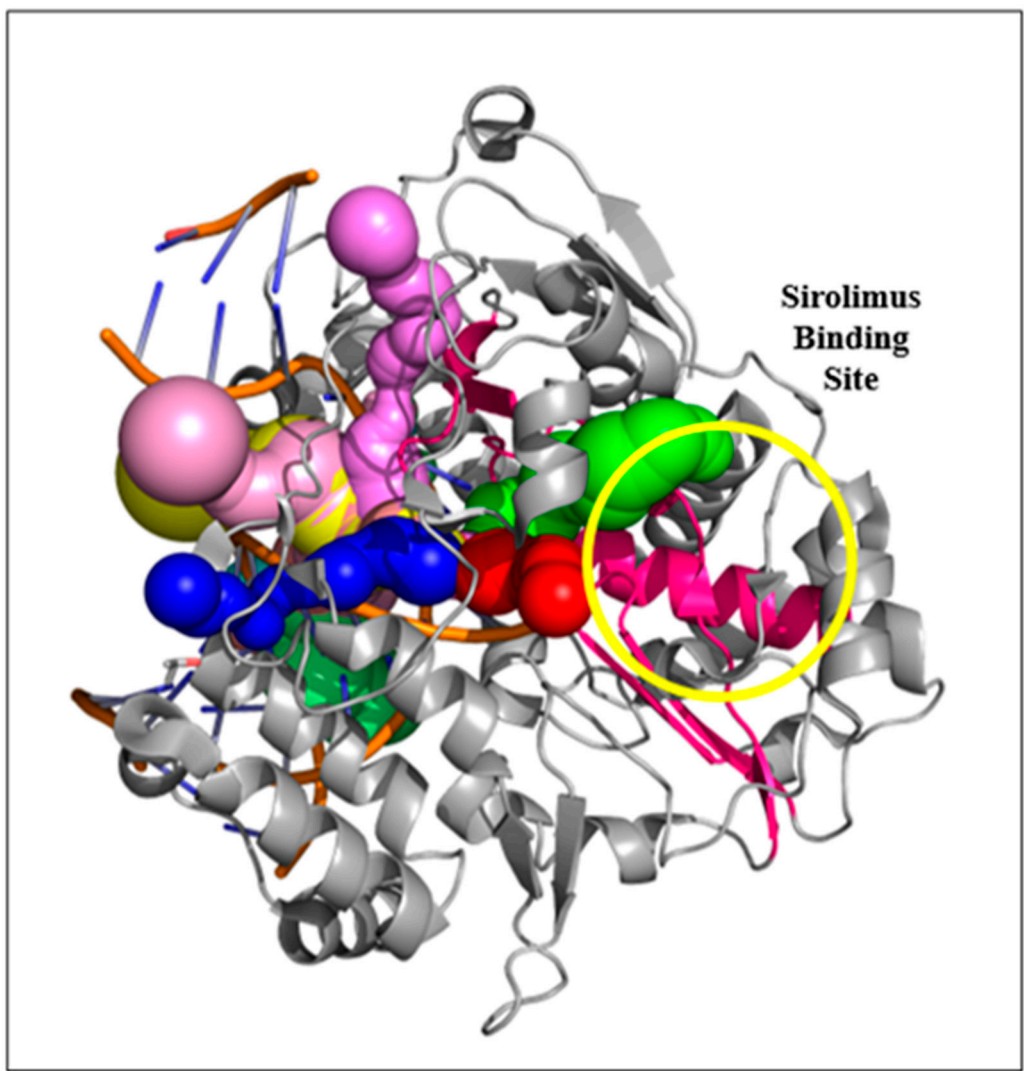

**Figure 13.** Tunnels in Poliovirus RdRp (PDB-ID: 2ijf). Vacant spaces were indicated by colored space filling balls for the template RNA (pink), newly synthesized RNA (blue), active site (red), NTP binding site (green). The sirolimus binding site (yellow circle) is at the NTP entry tunnel. Sirolimus binding blocks the entry point for NTP. Motifs (magenta) and RNA strands (orange) are shown. Tunnels were obtained from the PDBsum database [28].

The active form of SARS-CoV2 RdRp is a heteromeric polymer of RdRp (Nsp12) with Nsp7 and Nsp8. Sirolimus was docked to the complex (PDB ID: 7BV2) in order to determine if these proteins obstructed access to the binding site. Sirolimus still had access to its binding site at the NTP tunnel (Figure 14). RdRp also forms larger membrane-based RdRp replication complexes such as the 12-mer in Nodavirus (PDB 8FM9) [33,34] and coronaviruses [35]. We predict sirolimus will have access to its binding site in these crown-like structures.

Investigation of the sirolimus binding site started by examining the amino acids that interact with the drug. Lysine accounted for 55% of encounters (55/101) followed distantly by arginine (16) and aspartate (10). All hydrophobic and other amino acids accounted for only 19% of interactions.

The interactions of sirolimus with individual amino acids were assessed in one representative structure from each of the five virus families (Figure 15). Despite the low % sequence identities and divergent evolution, there was consistent overrepresentation of basic amino acids such as arginine and lysine in the binding pocket.

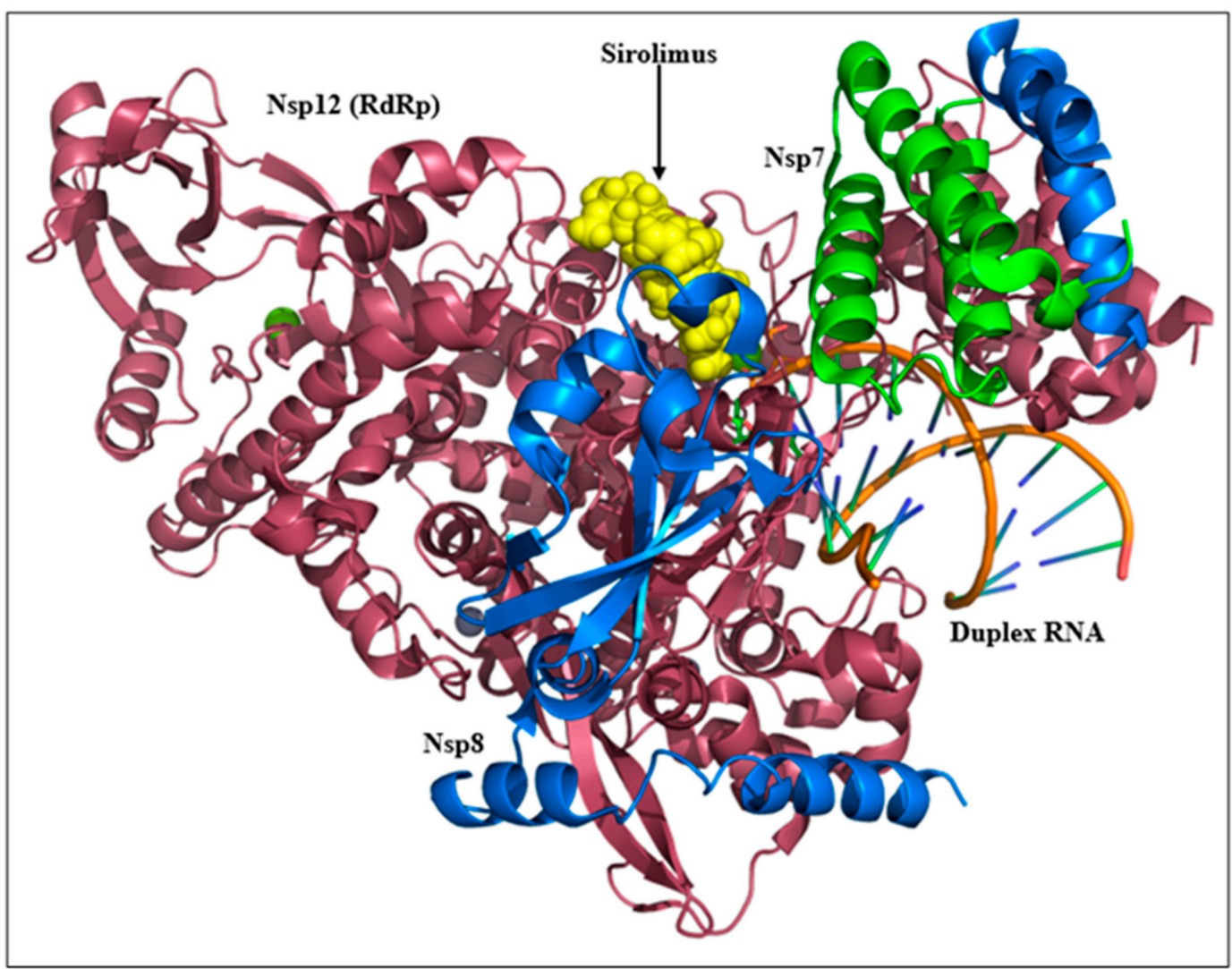

**Figure 14.** Sirolimus and active SARS-CoV2 RdRp complex (PDB ID: 7aap). The active complex consists of Nsp12 (RdRp, brown), the heterodimer of Nsp7 (green) and Nsp8 (blue), and a second monomer of nsp8 (blue). The sirolimus binding site at the NTP tunnel (yellow spheres) does not overlap with Nsp7, Nsp8 or RNA binding domains, or duplex RNA (orange backbones). This figure was created with the PyMol visualization software.

SARS-Cov-2 (PDB ID: 7bw4) bound sirolimus with 15 residues. These included the N-terminal region (D154, V156, E157) and finger (Y455). The two conserved strands of Motif F with K545 and R555 in the active site were not directly involved. However, charged amino acids in the loop that connected the two strands (Figure 4) contributed to the sirolimus binding site (K551, N552, R553). The loop in motif F had divergent sequences in each virus, but remained strongly charged with two to five lysine and/or arginine residues in the 24 viruses. Motif A and both of its conserved aspartates (D618, Y619, P620, K621 and D623) were involved. The NTP entry tunnel at the end of motif D (F793) and its continuing loop (S795, K798) completed the site. Consistent with Figure 13, sirolimus was bound to three lysine and three aspartate residues. Interactions with the N-terminal, the loop in motif F, motif D and NTP tunnel were outside the shared conserved RdRp motif.

Residues interacting with sirolimus are critical for nucleotide binding. The docking site was compared to the electrostatic surface (Figure 12) and a model of the nucleotide binding pocket [32]. Hydrogen bonds coordinate the nucleotide between T687 of motif B and K545 of motif F. The 3′-OH of ribose interacts with S682 of motif B and D623 from motif

A. The triphosphate and complexed $Mg^{2+}$ ions lie between oppositely charged plates in the active site and entry tunnel [32]. The two $Mg^{2+}$ are coordinated by negatively charged D760 and D761 in the active site of motif C and D618 of motif A. The phosphate string is stabilized by the positively charged basic plate formed by R553 and R555 of motif F, K621 in motif A and salt bridge with K798 that is distal to motif D. We propose sirolimus interacts with the alkaline plate of lysine and arginine residues to prevent anchoring of the triphosphate, ribose and nucleotide [32].

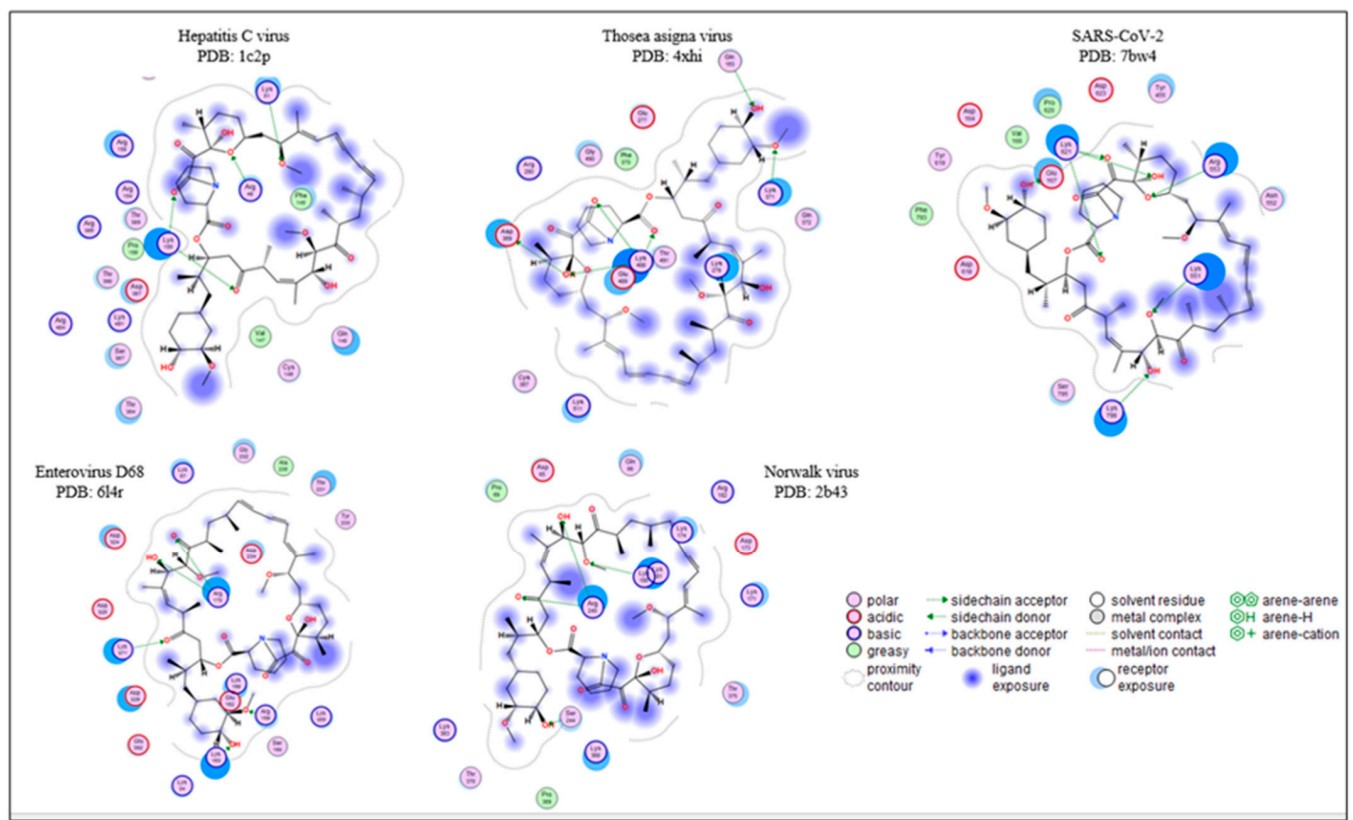

**Figure 15.** Sirolimus binding site. One representative species from each of the five virus families is selected to show the conservation of positively charged amino acids Arg and Lys residues that stabilize interactions with sirolimus.

Hepatitis C NS5b RdRp (PDB ID:1c2p) bound sirolimus with 18 amino acids beginning with the N-terminal (R48, K51). Motif F was extensively involved including conserved R158. Two sequences from the intervening loop between the two strands of motif F were unique to hepatitis C: F145 C146 V147 Q148 and R154 K155 P156. They maintained the basic electrostatic charge in the region of motif F. The alpha helix of motif D (T364, S367) and loop (R386, D387, T389 and T390) formed the lower lip of the NTP entry tunnel and sirolimus binding site. Other basic residues from the NTP tunnel were R484 and K491. Motifs A and B were not involved. Five of the amino acids were arginine, with three each of lysine and threonine.

There were 17 residues for enterovirus D68 (PDB 6l4r) beginning with K34 and K57 in the N-terminal. The active site in motif F contributed R170 while the loop residues R159, S160, E162, K163 and K168 also interacted with sirolimus. The active site of motif C (D324, D325) was contacted. The active site of motif A that interacts with the nucleotide ribose-3′-OH and pyrophosphate were involved (D229, T231, G232, Y233, D234 and A235). The sirolimus site included the approximate location of motif D and the NTP entry channel (K355, K371, E392). Lysine accounted for 6 residues with two arginines, four aspartate and two glutamic acids.

Norwalk virus RNA dependent RNA polymerase domain from strain Hu/NLV/ Dresden174/1997/GE (PDB 2b43) had 15 interactions with sirolimus. The N-terminal had a sequence of D66, Q67, K69 and P70. Motif F had the intervening sequence of K171, D173, K174 and active site R182. Motif A offered S244 and R245 that are between the conserved aspartate residues. The motif D and loop contributed K363, K368, P369, T370 and T375. Overall, there were five lysines, two aspartate and one arginine. Only nine amino acids were conserved between the proteins, but they were highly charged residues in the active site and common core that were concentrated in the electrostatic surfaces of motifs F, A and C and the NTP entry tunnel (Figure 5).

Motif F anchors the template ribose and nucleotide to the inner edge of the RNA tunnel under the fingers. Sirolimus was bound to the conserved arginine in motif F and basic residues in the intervening amino sequence of the loop (Figure 4). Although the structure of motif F was not always conserved in Flaviviridae, analogous lysine residues were found in comparable sequences of their RdRps. Motif B binds the template ribose but was not part of the sirolimus binding site. Motif A stabilizes the incoming NTP, ribose-3′-OH and metal ion. Conserved D618 and D623 in the active site were part of the sirolimus binding sites for SARS-CoV-2 and enterovirus D68 in addition to other residues of motif A. The pair of aspartates in the active site of Motif C were implicated for enterovirus D68. Motif D and its loop were poorly conserved structures that were not in the common core, but it did contribute charged residues to the NTP tunnel and were bound by sirolimus. These interactions explain the separation between remdesivir and sirolimus binding pockets (Figures 12 and 14) and predict that rapamycin will block the NTP entry tunnel and occupy the NTP pocket to interfere with primer extension during RNA replication.

## 3. Discussion

Our aim was to show that a structural biology approach can identify novel drug binding sites in RdRp. We started with the "common core" of ss(+)RNA virus RdRp structural motifs A to G identified by Peersen [10] that define the active site, but re-aligned them to define a shared conserved RdRp structure of motifs F, A, B and C. We bound FDA approved drugs to this highly conserved structure and identified sirolimus and other -limus drugs, rifampin antibiotics and digoxin as the top three candidates across a wide but representative array of viruses. An innovation was the discovery of a novel drug binding site in the NTP entry tunnel adjacent to the active site. The location was accessible in SARS-CoV-2 despite the presence of Nsp7, Nsp8, RNA and remdesivir. This approach is different from the majority of docking studies that have focused on individual virus species and have focused only on the active site in an attempt to interfere with motif C and RNA ligation.

Analysis of the sirolimus binding site found frequent hydrogen bond interactions with lysine residues of the electrostatic surface. Sirolimus interactions were predominantly with motif A, the intervening loop sequences in motif F and various residues in motif D and the NTP tunnel. The twin strand structure of motif F was not conserved in Flaviviridae, but the charged residues required for the active site and sirolimus binding were present in comparable positions and are likely to support drug binding. Structural modeling indicated sirolimus "capped" the NTP entry tunnel and thus may interfere with replication by blocking access of NTP to the active site. Other alternatives include steric hindrance or allosteric modulation of function. Motif B is on the opposite side from the NTP tunnel and was not part of the sirolimus binding pocket.

Remdesivir was developed to treat the negative sense single stranded RNA Ebola virus from family Filoviridae, and so is not an optimal nucleotide antagonist for ss(+)RNA viruses including SARS-CoV-2. The mediocre clinical outcomes and drug-induced hepatotoxicity may stem from shortcomings derived from optimistic extrapolation of in vitro testing results that were anticipated to readily translate to success in vivo [2]. Remdesivir had only one hit and no other nucleotide antagonist was selected.

A recent modeling study by Papalani et al. docked sirolimus, digoxin and other drugs into the active site of SARS-Cov-2 RdRp [17]. Both approaches found sirolimus binding to motifs F and A, but Papalani et al. also found binding to motif B, the active site aspartate dimer in motif C and residues in the thumb. Their approach did not use the entire enzyme and so could not identify binding to motif D and the NTP tunnel [17].

Sirolimus is a macrocyclic antibiotic that is used as an immunosuppressant to prevent organ rejection after kidney transplantation. Sirolimus targets the mTOR (mammalian target of rapamycin) pathway which is a central regulator of cellular metabolism. mTOR also plays a crucial role in autophagy and is a pharmacologic target for autophagy regulation [36,37]. The interplay between autophagic machinery and virus infections dates to the sixties [38]. Some viruses co-opt the mTOR pathway to effectively replicate in the host leading to mTOR suppression [39]. Suppression of the mTOR pathway leads to increased viral protein translation [40,41]. Not surprisingly, several efforts have therefore focused on the RdRp domain of the virus and the mTOR pathway as drug targets [41,42]. Sirolimus has been identified as one of the drugs against RdRp of many viruses and mTOR1 by several studies [17,18,39,40,43,44]. Therefore, sirolimus may have potential as a dual agent interfering with mTOR and RdRp during replication [45]. Sirolimus modulated PI3K/AKT/mTOR and ERK/MAPK signaling pathways of Middle East respiratory syndrome (MERS) coronavirus infection in vitro [46]. Sirolimus was identified as a potential drug for treating COVID-19 [17,47] and may have immunomodulatory properties as suggested by enhancement of COVID-19 vaccination in patients with β-thalassemia [48]. Because the conserved charged residues in the common core are essential for the activity of RdRp and are present in the drug target binding site, it is possible that sirolimus may remain effective despite amino acid substitutions caused by high viral mutation rates [44]. The congruence is evident by comparing the drug binding motif between species (Figure 16). Sirolimus is currently in clinical trials for treating COVID-19 (NCT04461340, NCT04341675, NCT04948203, https://clinicaltrials.gov, accessed on 1 December 2022). In addition to Sirolimus, Digoxin and Rifampin were in the top 10 hits (Supplementary Table S1). Cell-based assays have shown antiviral activity of these drugs in SARS-CoV-2 [49,50].

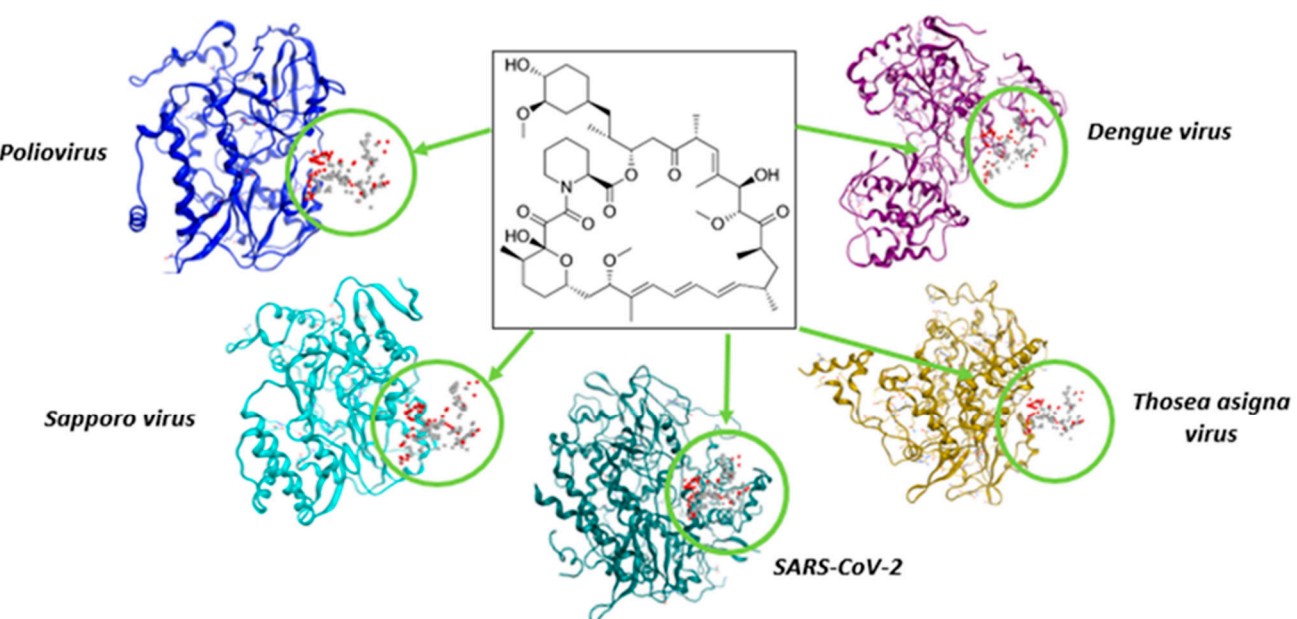

**Figure 16.** Sirolimus docked to the drug-binding site. One representative structure from each of the 5 virus families is shown. The green circle indicates the binding site.

Hepatitis C NS3 protease inhibitors asunaprevir and faldaprevir were the top antivirals identified from our Set 2 dockings [51,52]. NS3 is a serine protease required for proteolytic cleavage of the HCV polyprotein [51,53]. Asunaprevir inhibits SARS-CoV-2

propagation in vitro [54]. It is possible these sites share a common structural arrangement of electrostatic surfaces for sirolimus binding because polyanionic sulphated naphthy-lamine derivatives such as suramin and Evans Blue bind to positively charged patches of SARS-CoV-2 RdRp [55]. Suramin binds two sites and prevents binding of the RNA template and primer strands near the catalytic site [56]. The antibiotic quinupristin may bind the RNA tunnel to block access [18].

The results of this investigation support our hypothesis that an evolutionary conserved core can be used for drug discovery. This method is a novel approach to drug design that has not been pursued before. Future direction/s for this project include in vitro testing of the top ten drugs and antivirals and applying the same in silico drug design approach to other RNA or DNA virus families. The preservation of right hand structural similarity and conservation of motifs shown in Figures 3 and 4 indicate deep evolutionary relationships between monomeric polymerases of other RNA virus families, retroviruses and DNA viruses [57] and suggests that our approach may be valuable to identify broad-spectrum antivirals [58]. Structural and functional similarities extend to the replicative enzymes of eukaryotic mobile genetic elements such as LTR- and non-TLR retrotransposons and telomere extension polymerases [59], group II introns [60], pre-mRNA-splicing factor 8 in the eukaryotic spliceosome [61] and diversity-generating retroelements (DGR) in prokaryotes [62] that are members of the superfamily of monomeric DNA- and RNA polymerases with fingers, palm and thumb functional subdomains [63]. Interactions at these sites should be examined to anticipate potential drug toxicities and to improve antiviral specificity.

This study was limited to 24 species from 5 different families of ss(+)RNA viruses. The modeling will need to be verified by crystallography and protein binding studies with the top candidate drugs and purified RdRp enzymes. The common core approach may be validated by studying structures from additional virus families and can now be applied to other monomeric polymerases that share the highly conserved motifs. This study is based on a targeted set of FDA approved drugs to demonstrate proof of concept. In the future, this approach can be expanded to other 900 million compounds available in public databases such as PubChem (https://pubchem.ncbi.nlm.nih.gov/) and ZINC (https://zinc.docking.org/). Docking of millions of compounds is computationally expensive and time consuming.

Our result can be compared to current efforts of drug development [64–66]. Nucleotide triphosphate analogues are effective inhibitors of RdRp, but drug delivery is limited by rapid metabolism of the triphosphate. Instead, drugs such as sofosbuvir that was developed for dengue virus are chemically modified and become active for chain termination after being metabolized [67]. Set 2 NTP analogues and other antiviral drugs had poor binding to our drug binding motif (Figure 9).

Non-nucleoside analogue inhibitors are noncompetitive allosteric inhibitors that bind to thumb pockets I (T1) and II (T2) and palm pockets I (P1), II (P2) and β (Pβ) that are adjacent to the active site [12]. These drugs prevent conformational changes required catalytic activity. Drugs active at T1 and T2 pockets prevent assembly of appropriate palm and thumb interactions and decrease binding of the RNA template. Inhibitors that fit in the hydrophobic palm P1 and P2 pockets stabilize an inactive conformation of the enzyme and inhibit phosphodiester bond formation. Hepatitis C has been extensively studied [29] but non-nucleoside analogue inhibitors that act allosterically outside the RdRp active site [30] were not identified in our drug searches.

Additional drugs target protein–protein interactions of RdRp. The N-terminal domain of SARS-CoV2 RdRp curls back and its edge contributes to the sirolimus binding site in COV2 (Figure 16). The N-terminal methyltransferase domain of dengue virus (PDB ID: 4hhj) also curls back over the enzyme to form the active conformation [68]. In addition, dengue protease (NS3) must bind to RdRp (NS5) for replication to proceed. Drugs that block the protein interaction inhibit dengue replication [69]. Host translation elongation

factors EF-Tu and EF-Ts and the S1 ribosomal protein interact with Qβ polymerase (PDB ID: 4r71) and represent an additional family of drug targets [70].

Targeted covalent inhibitors (TCIs) are a recent innovation to design drugs that bind with specifically to critical side chains in enzymes to block them in reversible or nonreversible covalent fashion.

It is possible that peptide nucleic acid (PNA) or other oligomers that act as artificial nucleic acids may also serve as inhibitors by binding template RNA sequences or other mechanisms [71].

The wide diversity of RdRp structures outside the conserved motifs and channels and extensive range of idiosyncratic viral and host protein binding patterns offer many opportunities for virus specific drug development. This approach is in contrast to our aim of searching for highly conserved drug binding site(s). Nucleotide analogues, nonnucleotide inhibitors, protein interaction blockers and targeted covalent inhibitors did not interact in the region of the NTP entry tunnel as illustrated by SARS-CoV2 (Figure 16) and other RdRp (Figure 13). Our approach of searching for a widely conserved drug binding motif led to identification of a novel inhibitory site that may have the potential to lead to development of broad-spectrum antivirals that obstruct the NTP tunnel. Future stages of drug development will proceed by in vitro drug RdRp binding and inhibition studies, mutagenesis of the binding region, drug bound structure analysis, cell and animal testing with active viral infections.

## 4. Conclusions

We hypothesized that in silico structural biology approaches can discover novel drug binding sites for RNA-dependent-RNA-polymerases of positive single strand RNA virus species. In silico methods tested this hypothesis and identified sirolimus as a drug that would bind to a novel epitope extending from N-terminal domain to basic residues from motifs F, A and distal to D that form the NTP entry tunnel. This novel drug target and platforms of limus, rifampin and digitalis drugs may generate new broad-spectrum antivirals directed at RdRp of these viruses. A broad spectrum antiviral may be beneficial for cases and epidemics of ss(+)RNA viruses that are unknown, emerging or have no established therapies. Such a drug could act as a "viral penicillin" directed at the structurally conserved motif, but with the caveat that over time the viruses would evolve to display some resistance.

## 5. Materials and Methods

Identification of a common core: Peersen performed a comprehensive super positioning of viral polymerases from 414 protein data bank (PDB) entries [10]. The collection can be used to compare structures by loading PDB files into a molecular graphics program. The complete set of superimposed polymerase coordinates are publicly available in the community site Polymerase Structures at Zenodo, the Open Science platform at the CERN Data Centre The Zenodo publicly available database (https://www.zenodo.org/communities/pols/?page=1&size=20, accessed on 1 November 2020) was used to align three-dimensional RdRp structures from 24 ss(+)RNA viruses across five virus families (Table 1).

Drug docking process: The 24 aligned RdRp structures were loaded to Molecular Operating Environment (MOE) docking 2020.09 [72] (Table 1) and each structure individually minimized using the QuickPrep function. The site finder function was used to locate a site for docking near the common core of the RdRp. The site finder identified two distinct binding sites. One overlapped with the active site commonly used for drug docking by most studies in the literature. The second site was the new drug binding site. Site finder automatically found a single site shared by all 24 RdRp structures. Analysis of the amino acids from each of the 24 structures confirmed that the site was shared by all the structures and formed a new binding site distal to the active site. The dock routine within MOE was used to dock the 833 drugs from Set 1 to the drug binding site. The algorithm uses gradient-based conformational searches and an empirical scoring function based on the interaction of the ligand with the protein. MOE is very efficient at docking drugs based on several

physiochemical properties. Three-dimensional protonation and energy minimization were selected in MOE until a gradient of 0.05 was reached. Polar hydrogens were added. A grid of 1 Angstrom and a maximum of 30 binding modes were selected. The docked drugs were ranked based on their docking scores (S, kcal/mol), and the top ten highest affinity drugs for each protein were selected for evaluation of molecular interactions. The same procedure was repeated for the docking of Set 2 antiviral drugs. While we were limited by the availability of solved 3D structures, each structure was minimized and docking was performed individually to remove any modeling bias that may exist in very similar structures. Since the process of docking requires extensive computational power, we were limited by how much computational space we were able to search.

Set 1 contained 1993 FDA-approved drugs in the e-drug 3D database in SDF format [73]. Drugs were filtered to minimize the conformational search space and selected 833 drugs with ≤8 rotatable bonds. Because the e-drug database had an incomplete list of antivirals, Set 2 of 85 antivirals were downloaded with DrugBank with permission [74]. Drugs were docked to the new site using MOE [72].

Structural Visualization: PyMOL visualization software (www.Pymol.org, accessed on 1 April 2021) was used to create all the figures in this manuscript.

Visual Molecular Dynamics (VMD 1.9.3): VMD was used to align the amino acid sequences of the 24 common cores using the MultiSeq routine and Stamp Structural Alignment tool. Structural homology of the alignment is provided by $Q_H$ and Root Mean Squared Deviation (RMSD) values. $Q_H$ and RMSD values reflect the overall structural conservation between proteins. $Q_H = 1$ when structures are identical. The Phylogenetic feature within MultiSeq was used to construct structure-based trees based on $Q_H$ and RMSD.

Sequence alignment: Multiple sequence alignment (MSA) of the common core was performed using the online Muscle Sequence Alignment tool (https://www.ebi.ac.uk/Tools/msa/muscle/, accessed on accessed on 21 May 2022).

**Supplementary Materials:** The following supporting information can be downloaded at: https://www.mdpi.com/article/10.3390/biomedinformatics3040055/s1, Supplementary Table S1. Top 10 drugs for each virus and docking parameters. Supplementary Table S2. Antiviral drug docking parameters.

**Author Contributions:** A.S.G. conceived the idea, performed the docking analyses, created all the figures and wrote up the first version of the manuscript. J.N.B. mentored and guided A.S.G., contributed to discussions on various aspects of the work and to the editing and writing-up of the manuscript. All authors have read and agreed to the published version of the manuscript.

**Funding:** Jerry Petrak Research Fund.

**Institutional Review Board Statement:** No review necessary.

**Informed Consent Statement:** No informed consent necessary.

**Data Availability Statement:** Data are freely available from A.S.G. upon e-mail request.

**Acknowledgments:** A.S.G. thanks S. Vasudevan, Professor, Georgetown University, for being a great mentor, and for providing access to MOE, the software that was used for the docking studies performed herein.

**Conflicts of Interest:** Authors declare no conflict of interest.

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
