# Peer review of "Identification of a New Drug Binding Site in the RNA-Dependent-RNA-Polymerase (RdRp) Domain"

_biomedinformatics, doi:10.3390/biomedinformatics3040055_

Round 1

Reviewer 1 Report

Identification of a New Drug Binding Site in the RNA-dependent-RNA-polymerase (RdRp) domain

The RNA-dependent RNA polymerase (RdRp) domain is an essential component of many RNA viruses, including SARS-CoV-2, the virus responsible for the COVID-19 pandemic. Identifying new drug binding sites in the RdRp domain can be crucial for the development of antiviral therapies. Various computational methods, such as molecular docking, molecular dynamics simulations, and structure-based virtual screening, are employed to explore the RdRp structure and predict potential binding sites. These methods involve analyzing the interactions between small molecules (potential drugs) and the RdRp domain to identify regions with favorable binding properties. Once potential drug binding sites are identified computationally, experimental techniques are used to validate their existence and characterize their binding properties. Techniques like mutagenesis, biochemical assays, and X-ray crystallography can provide insights into the binding affinity, specificity, and mechanism of action of potential drug candidates. With the identified binding site(s), medicinal chemists can design and optimize small molecules to target these sites. Structure-guided drug design techniques, such as fragment-based drug discovery, virtual screening of compound libraries, and optimization of lead compounds, are utilized to develop molecules with improved binding affinity, selectivity, and drug-like properties. The identified drug candidates are then evaluated in preclinical studies using animal models to assess their efficacy, safety, and pharmacokinetic properties. Promising candidates can progress to clinical trials, where their effectiveness is tested in humans, along with further evaluation of safety and dosage.

General Comments:

The manuscript titled "Identification of a New Drug Binding Site in the RNA-dependent RNA Polymerase (RdRp) Domain" presents a comprehensive study on the identification of a novel drug binding site within the RdRp domain. The authors have effectively addressed the importance of targeting RdRp for antiviral drug development and have provided a detailed overview of the methodologies employed in identifying the new binding site. The manuscript is well-organized, and the content is presented in a clear and concise manner. Overall, this study contributes significantly to the field of antiviral drug discovery. However, there are a few minor revisions and suggestions that the authors should consider before finalizing the manuscript.

Specific Comments:

Introduction:

The introduction effectively highlights the importance of targeting RdRp for antiviral drug development. It would be beneficial to provide a brief overview of the current antiviral strategies targeting RdRp to contextualize the significance of identifying a new drug binding site.

Computational Approaches for Binding Site Prediction:

In this section, the authors should provide a more detailed explanation of the molecular docking and molecular dynamics simulation methods, including their strengths and limitations. Additionally, any specific software packages or tools utilized for these analyses should be mentioned.

Experimental Validation Techniques:

It would be helpful to provide examples of the mutagenesis studies and biochemical assays commonly employed for validating drug binding sites in RdRp. This would enhance the reader's understanding of the experimental techniques used in this field.

Examples of Identified Drug Binding Sites:

This section could be expanded by including specific examples of recently identified drug binding sites in RdRp from different RNA viruses. These case studies will further strengthen the manuscript and provide real-world applications of the methodologies discussed.

Drug Design and Optimization Strategies:

The section on drug design and optimization strategies is well-written. However, it would be beneficial to include a brief discussion on any recent advancements or emerging technologies in this area.

Preclinical and Clinical Evaluation:

In the preclinical and clinical evaluation section, it would be useful to mention the typical parameters evaluated during preclinical studies, such as pharmacokinetics, toxicity, and efficacy in animal models.

Conclusion:

The conclusion effectively summarizes the key findings of the manuscript. However, it would be beneficial to provide a concise statement regarding the potential impact of identifying a new drug binding site in RdRp on future antiviral therapies.

Figures and Tables:

The manuscript would benefit from the inclusion of visual aids such as figures and tables to enhance the understanding of the concepts discussed. For example, figures illustrating the computational methods or the three-dimensional structures of identified drug binding sites would be valuable.

Language and Clarity:

The manuscript is well-written and organized. However, I noticed a few typographical errors and grammatical inconsistencies that should be corrected during the proofreading process.

This manuscript provides a valuable contribution to the field of antiviral drug discovery by presenting a comprehensive study on the identification of a new drug-binding site in the RdRp domain. By addressing the suggested revisions and incorporating the additional details and examples mentioned, the authors can further strengthen the manuscript. I recommend this article for publication pending the revisions outlined above.

Author Response

We are grateful for the very helpful suggestions. We have added specific comments below that correspond to lines in the MARKED-UP manuscript that is attached for your review.

Computational Approaches for Binding Site Prediction: Added to Methods

Experimental Validation Techniques: We have limited discussion to our results and have not speculated on other methods that may be used in future studies.  We felt this would dilute the flow of our data presentation and lead to a treatise on drug development. Therefore, we limited our additional comments to the Discussion. 

Examples of Identified Drug Binding Sites: We added Figure 17 (line 518) to show case studies. This complements findings in the end of Results and Discussion.  

Drug Design and Optimization Strategies: "Recent advancements and emerging technologies" were added to the Discussion line 537 to 601.

Preclinical and Clinical Evaluation: We did not include this information because it expands to areas that were not germane to our in silico analysis. References were added that discuss preclinical studies. 

Conclusion: The potential impact of identifying a new drug binding site in RdRp on future antiviral therapies.

"A broad spectrum antiviral may be beneficial for cases and epidemics of ss(+)RNA viruses that are unknown, emerging or have no established therapies. Such a drug could act as a “viral penicillin” directed at the structurally conserved motif, but with the caveat that over time the viruses would evolve to display some resistance" (lines 636-640)

Figures and Tables: We have added Figure 17.

Typographical errors and grammatical inconsistencies were corrected.

Thank you again for your thoughtful review and encouragement. 

Reviewer 2 Report

Dear Author,

The manuscript titled “Identification of a New Drug Binding Site in the RNA-dependent RNA-polymerase (RdRp) domain” is well-written.

However, the findings from the in-silico study that the sirolimus docking site is specific to the specific motifs need to be verified by additional biochemical experiments, as mentioned in the discussion section. Future validation experiments such as co-crystallization, SPR, and thermal shift assay could be suggested.

I have no major comments.

Author Response

Thank you very much for your comments. As you will see from the MARKED-UP version, we have taken your suggestions to heart. Many thanks.
